# CMPC: An Innovative Lidar-Based Method to Estimate Tree Canopy Meshing-Profile Volumes for Orchard Target-Oriented Spray

**DOI:** 10.3390/s21124252

**Published:** 2021-06-21

**Authors:** Chenchen Gu, Changyuan Zhai, Xiu Wang, Songlin Wang

**Affiliations:** 1Beijing Research Center of Intelligent Equipment for Agriculture, Beijing 100097, China; guchenchen57@nwafu.edu.cn (C.G.); wangx@nercita.org.cn (X.W.); 2National Engineering Research Center of Intelligent Equipment for Agriculture, Beijing 100097, China; 3College of Mechanical and Electronic Engineering, Northwest A&F University, Yangling 712100, China; 4College of Mechanical Engineering and Automation, Liaoning University of Technology, Jinzhou 121001, China; jxwangsl@lnut.edu.cn

**Keywords:** orchard tree, LiDAR, pesticide, point cloud, target-oriented spray

## Abstract

Canopy characterization detection is essential for target-oriented spray, which minimizes pesticide residues in fruits, pesticide wastage, and pollution. In this study, a novel canopy meshing-profile characterization (CMPC) method based on light detection and ranging (LiDAR)point-cloud data was designed for high-precision canopy volume calculations. First, the accuracy and viability of this method were tested using a simulated canopy. The results show that the CMPC method can accurately characterize the 3D profiles of the simulated canopy. These simulated canopy profiles were similar to those obtained from manual measurements, and the measured canopy volume achieved an accuracy of 93.3%. Second, the feasibility of the method was verified by a field experiment where the canopy 3D stereogram and cross-sectional profiles were obtained via CMPC. The results show that the 3D stereogram exhibited a high degree of similarity with the tree canopy, although there were some differences at the edges, where the canopy was sparse. The CMPC-derived cross-sectional profiles matched the manually measured results well. The CMPC method achieved an accuracy of 96.3% when the tree canopy was detected by LiDAR at a moving speed of 1.2 m/s. The accuracy of the LiDAR system was virtually unchanged when the moving speeds was reduced to 1 m/s. No detection lag was observed when comparing the start and end positions of the cross-section. Different CMPC grid sizes were also evaluated. Small grid sizes (0.01 m × 0.01 m and 0.025 m × 0.025 m) were suitable for characterizing the finer details of a canopy, whereas grid sizes of 0.1 m × 0.1 m or larger can be used for characterizing its overall profile and volume. The results of this study can be used as a technical reference for the development of a LiDAR-based target-oriented spray system.

## 1. Introduction

Pesticide application is necessary for controlling pests and maintaining fruit quality in orchards. Continuous and undifferentiated pesticide applications are often performed in orchards. However, there are gaps between trees, and their canopies are generally non-continuous. This causes pesticide solution wastage, environmental pollution in orchards, and an increase in pesticide residues in fruits. Targeted spray only applies pesticide when the target is present and can reduce pesticide wastage and pollution [1,2,3]. However, this approach does not allow pesticide delivery to be varied according to the characteristics of tree canopies. The targeted variable-rate spray allows pesticide usage to be precisely varied according to the target characteristics. In this case, the sprayer is fitted with sensors that measure the canopy characteristics and enable the correct pesticide dosage to be applied. This technology can considerably reduce the amount of pesticide application and waste, reduce agricultural investment costs, and increase the income of farmers [4]. Canopy detection is an important part of the variable-rate spray method. The key canopy characteristics for variable pesticide application include canopy position, profile, volume, and biomass, which can be obtained using a variety of sensors [1,2,3,5,6,7]. Infrared photoelectric detection, microwave detection and image detection technologies have been used for canopy detection [8]. However, infrared photoelectric detection technology can only detect the presence or absence of trees, and cannot set the spray volume [8]. Microwave detection is limited by microwave propagation characteristics, complex control, and poor economy [9]. Meanwhile, image detection technology is hindered by complexity, poor stability, a large amount of data to be processed, slow response speed and high cost [10]. Following the rapid development of sensor technologies, various types of sensors have been successfully applied to canopy measurement [11,12,13,14,15,16,17]. Ultrasonic and light detection and ranging (LiDAR) sensors are currently the most popular approaches for characterizing orchard-tree canopies.

Ultrasonic sensors for variable spraying emit and receive ultrasonic signals to measure orchard-tree canopy thicknesses [11,12,18,19,20,21] in real time. Variable-rate spray systems in combination with ultrasonic sensors can reduce pesticide usage by 34–58% compared with conventional spraying techniques [1,13,14,22]. Furthermore, the target-detection accuracies of these ultrasonic systems generally exceed 87% [14,23]. However, the accuracy is affected by the detect on distance, beam angle [24], and the speed of the moving sensor. [25]. LiDAR sensors are fast, accurate, insensitive to environmental conditions [26], and can be used for target characterization in variable pesticide applications [27,28,29]. Llorens et al. [30], Qiu et al. [31], and Tumbo et al. [16] compared the performance of ultrasonic and LiDAR sensors in canopy characterization. Their results showed that the ultrasonic sensor was best suited for determining the averaged canopy features, whereas LiDAR was better suited for extracting precise information about the canopy. LiDAR systems emit and receive high-frequency laser signals, calculating distances through the time-of-flight principle. Distances between the LiDAR system and the outer edges of the orchard tree canopy are obtained by varying the scanning angle of the laser such that a point cloud of the canopy is produced. The characteristics of the target canopy can then be obtained by processing the point-cloud [28,32,33]. Lee et al. [34] used the convex hull method to process point-cloud data. They obtained single-sided 2D profiles of the target tree using longitudinal scans and then calculated the canopy volume by assuming the crown to be symmetric with respect to a vertical line on the tree trunk. Yu et al. [3] used a slicing method to obtain the point cloud of an entire orchard tree by moving the LiDAR system around each orchard tree in a circle. The slices were constructed at equidistant intervals along the height of the crown. The canopy volume was calculated by multiplying the number of slices by the length of intervals. The abovementioned methods are accurate for trees with symmetric crowns (e.g., citrus trees) but unsuitable for trees with asymmetric crowns (e.g., apple and pear trees), and thus have limited applicability. Li and Chen treated tree canopies equivalent to multiple regular, different-sized objects [5,35], and the canopy volume was calculated by adding the volumes of regular objects. Cai et al. [36] employed variable grid sizes to improve the accuracy of LiDAR-based canopy volume measurements. These methods calculating canopy volume are suitable for asymmetric fruit-tree canopies and can be adapted to various tree canopies. However, these methods require a large number of computations with a slow operating speed. Further, the point data obtained were poorly utilized and could not capture the entire geometric features of the canopy. The sizes of objects and grids cannot be accurately determined for target-oriented spraying, which limiting their applicability to practical problems. 

The orchard-tree canopy profile can represent the outer edge and shape of the tree. An extensive set of canopy features such as volume, geometric characteristics, and shape are included [18,25]. Studies on the computation of canopy profiles are scarce and there are few reports on the influence of measurement parameters. Zhai et al. [18,21] succeeded in measuring orchard-tree canopy profiles using ultrasonic sensors. The canopy volume detection accuracies were 87% and 90% for hawthorn and cherry trees, respectively. Ultrasonic waves feature a beam angle; thus, to reduce interference between ultrasonic signals, the number and layout of the ultrasonic sensors should be carefully optimized. However, the detection accuracy can be affected by the number of ultrasonic sensors. LiDAR can detect the canopy with high accuracy with smaller angular resolutions. 

The objective of this research was to develop an innovative detection method for calculating the canopy volume for a target-oriented spray with high precision. A canopy meshing-profile characterization (CMPC) method based on LiDAR point-cloud data was designed. Both laboratory and field experiments were carried out to evaluate the accuracy of CMPC. Meanwhile, the effects of the LiDAR detection speed and algorithm grid size on CMPC performance were also investigated. The results were further compared with previous results obtained using ultrasonic sensors. The findings of this study can serve as a technical reference for the application and development of target-oriented spraying.

The remainder of this paper is organized as follows. Section 2 describes the operation of the two data acquisition systems, LiDAR-based CMPC, and manual measurement. Details on the design of the CMPC and data processing methods are also presented. Section 3 presents the results of the laboratory and field test experiments. Section 4 discusses the effects of the moving speed and algorithm grid size on the CMPC canopy measurement performance. Finally, Section 5 provides the conclusions of the study.

## 2. Materials and Methods

### 2.1. Data Acquisition Systems

#### 2.1.1. LiDAR Subsection

A sliding LiDAR measurement system was designed and constructed for this study. It consisted of a notebook computer (Lenovo T470p), stepper motor controller (KH-01), stepper motor driver (HB860D), stepper motor (86HB250-80B), linear rail guide (CCM-W60-35), slider, and sliding platform. The stepper motor controller instructed the stepper motor to rotate at the set speed. The slider could move along the linear rail guide. The typical operating speed of an orchard pesticide sprayer should below 1.5 m/s [37] for better spray performance. The maximum moving speed of the slider were set as 1.2 m/s. The moving range of the slider was 0–5 m, which allows detection of the whole tree canopy. The slider was situated 1.5 m above the ground. The LiDAR sensor was mounted on the slider and moved therewith along the linear rail guide. The structure of the sliding LiDAR system is shown in Figure 1. A Sick LMS111-10100 2D LiDAR sensor (SICK AG, Waldkirch, Germany) was used in this system. The scanning rate was 50 Hz. The angular resolution was 0.5° and the scanning range was −45°–225°. Each LiDAR data series contained 541 data points, and the data resolution was 10 mm. To maximize the data transfer rates, the notebook computer was connected to the LiDAR sensor via a cable interface. The point-cloud data were stored using SOPAS software, and a.txt file containing hexadecimal output data was produced at the end of each measurement.

#### 2.1.2. Manual Measurement System

A manual measurement system was designed for measuring orchard tree canopies (Figure 2). Mounting plates were installed on a sliding platform using instrument mounting brackets. An infrared-laser rangefinder (LM40) and an infrared-laser digital spirit level (EM5416-150) were fitted on the mounting plate to measure the distances to the canopy profile and to level the mounting plate, respectively.

### 2.2. Canopy Meshing-Profile Characterization (CMPC) Method and Data Processing Method

#### 2.2.1. CMPC Method

In the grid coordinate system, y is the horizontal direction toward the LiDAR sensor, z is the height, and x is the canopy-thickness direction. The grid intersection coordinates of the canopy were obtained by meshing the canopy in 0.1 m intervals in the horizontal (y) and longitudinal (z) directions from the bottom-most point of the LiDAR point cloud. An area was delineated around each (y, z) grid intersection point, at 0.5 × grid size in the up, down, left, and right directions, and the maximum x value in this area (x_max_; i.e., the thickness of the canopy) was set as the x coordinate of the grid intersection point. The (x, y, z) coordinates of the canopy profile were thereby obtained. A 3D representation of the canopy profile was constructed, and the canopy parameters were computed and extracted from this 3D profile. The meshing algorithm for computing the canopy profile is illustrated in Figure 3.

#### 2.2.2. Point-Cloud Density of Orchard Tree Canopy

The point-cloud density is an important factor that determines whether the shape, structure, and position of a subject can be accurately extracted from the point cloud. A set of canopy point-cloud data was extracted from the LiDAR measurements at intervals of 1 mm under a moving speed of 0.05 m/s. These measurements were processed and then compared against the manual measurement results. A schematic illustration of the point-cloud calculations is presented in Figure 4.

Each top-to-bottom scan of the orchard tree required 130 points/set to be calculated. 92 points/set were required for each scan of the canopy. LiDAR scanning the entire tree and crown required 325,000 and 230,000 points, respectively, and generated 2500 point-cloud datasets. The orchard tree point-cloud density was 52,000 points/m^2^, whereas that of the canopy was 55,100 points/m^2^. These point-cloud densities exceed those used by Escolà et al. [24] (8000 points/m^2^) and Colaço et al. [28] (700 points/m^2^). Therefore, the profile and position of the orchard tree can be accurately determined from our point-cloud data.

#### 2.2.3. Point-Cloud Data Processing

The original point cloud data obtained from LiDAR scans of the target tree were extracted using MATLAB. The data were sorted using scanning angles of −45–225° in 0.5° increments by converting polar coordinates to Cartesian coordinates [28,38,39]. The (x) values of the points that exceeded the horizontal distance between the LiDAR device the and tree trunk (2 m) were excluded from the dataset. The pesticide application was typically performed on one side of the canopy of an orchard tree. Therefore, only the point-cloud data that corresponded to one side of the orchard tree were considered, and the (x, y, z) coordinates for these data were defined. The LiDAR data were then compared against manually measured data to verify the feasibility of the LiDAR method, and the effects of each measurement variable on the computational results were investigated.

#### 2.2.4. Calculation of Canopy Characteristic Parameters

Each horizontal and longitudinal cross-section corresponded to a different height and horizontal position, respectively. The canopy volume could be obtained by summing the multiplicative products of each horizontal (longitudinal) cross-sectional area and its corresponding height (width). The maximum z-value in the vertical direction was treated as the canopy height, whereas the differences between the start and end points (outer fringes) of the canopy in the horizontal direction (range of y values) was treated as the canopy width. Figure 5 illustrates how a horizontal cross-section was used to calculate the procedure for calculating the canopy cross-sectional area. First, points on the horizontal cross-section profile were combined to form a line (i.e., one of the canopy horizontal profiles). Then, points corresponding to a profile thickness of zero were removed, and the area between adjacent points was calculated using the trapezoid area formula. After this, the areas were summed (Figure 5). The mathematical expression for this process is given by:(1)$$S_{\mathrm{section}}=\sum_{i=1}^{n}\frac{\left(x_{k}{+x}_{\left(k-1\right)}\right)}{2}(y_{k}-y_{\left(k-1\right)})$$
where, *S*_section_ is the horizontal area (m^2^), x*_k_* and x _(*k*–1)_ are the canopy thicknesses (m), and *y* _(*k*)_ and *y* _(*k*−1)_ are the horizontal distances (m).

The canopy volume is calculated as follows,
(2)$$Volume_{\mathrm{canopy}}=\sum_{i=1}^{n}S_{\mathrm{section}}\times d_{\mathrm{section}}$$
where, *Volume*_canopy_ is the canopy volume (m^3^), *S*_section_ is the horizontal area (m^2^), d_section_ is the profile interval (m). 

### 2.3. Testing and Validation of CMPC Method 

A simulated canopy model was constructed to test the viability and accuracy of the proposed CMPC method [40]. The canopy parameters of the simulated tree and an orchard tree were accurately measured using the manual method for comparative analysis. During manual measurements, the canopy thickness at each point on the canopy profile was obtained by positioning the receiving plate, which received the signal of the rangefinder, on the outermost edge of the canopy (as stably as possible).

#### 2.3.1. Experiments with Simulated Canopy in the Laboratory

The simulated canopy was constructed by stacking 30 cm × 20 cm × 10 cm cardboard boxes. The dimensions are shown in Figure 6. In the experiment, the LiDAR system was set 2 m away from the centerline of the simulated canopy.

#### 2.3.2. Target Orchard Tree 

The orchard experiment was conducted at the Beijing Xiaotangshan Modern Agricultural Science and Technology Demonstration Park (longitude: 160° 26.695′, latitude: 40° 10.806′). The actual orchard tree characterized in this experiment is shown in Figure 7. The target tree was 2.5 m tall and its canopy was 0.83 m above the ground. The height and width of the canopy are 1.67 m and 2.5 m, respectively. The experiment was conducted on 23 August 2019, on a cloudy day with temperatures of 19–32 °C and Class 1–2 south-easterly winds.

#### 2.3.3. Field Experiment

The distance between the LiDAR system and the centerline of the orchard tree was set to 2 m. The LiDAR sensor was moved back and forth along a predefined distance at a predefined speed to obtain the 3D point cloud of the canopy. Three measurements were taken for each moving speed. The process was repeated three times for each detection speed. The data were obtained and stored. The resulting data files (raw LiDAR data) were then exported in MATLAB for data processing.

In the manual measurements, the mounting bracket was moved 0.1 m along the rail guide after each measurement, while the mounting plate was moved upwards by 0.1 m (from an initial elevation of 0.83 m). An infrared receiving plate was used to measure the maximum canopy thickness at each position. To minimize human error during these experiments, the rangefinder was set to continuous measurement mode and automatically displayed the measured data. The measurements were then manually read from the display and recorded. During this experiment, a circular receiving plate was used to receive the infrared beams whose surface area was slightly smaller than the grid size of the algorithm. This made the plate easier to direct and control but with a slight loss of accuracy.

#### 2.3.4. Validation of CMPC Method

The operating speed is an important factor for both the operational efficiency of spraying machines, and the distance between LiDAR datasets with different point-cloud densities. Therefore, the study of the effects of moving speed on the accuracy of LiDAR-based canopy profile measurements was important. To this end, the canopy profile was measured using LiDAR at a variety of moving speeds (0.05 m/s, 0.2 m/s, 0.4 m/s, 0.6 m/s, 0.8 m/s, 1.0 m/s, and 1.2 m/s).

The effects of grid size on the canopy profiles obtained with the CMPC method were investigated. Accuracy tests were performed with different grid sizes, using experimental data collected at a moving speed of 1 m/s. The grid sizes selected for this analysis were factors of 0.1 m × 0.1 m; that is, 0.01 m, 0.025 m, and 0.05 m. Expressed otherwise, one cross-section was selected from every 10, 4, and 2 cross-sections to be compared with a manually measured cross-section at the same position.

## 3. Results

### 3.1. Analysis of LiDAR Measurements Errors Due to the Sliding Platform

In the simulated canopy experiment, the differences between the point-cloud datasets varied according to the LiDAR moving speed. The distance traveled by the LiDAR sensor on the sliding platform was susceptible to measurement errors. To measure and eliminate these errors, signboards were installed at the start and end points (5 m apart) of the slider′s range of travel. The errors between the relative positions of the LiDAR sensor on the sliding platform and the actual distance traveled by the LiDAR sensor were measured. The relative errors of the sliding LiDAR measurement system are shown in Table 1.

The measurement error depended on the moving speed and its magnitude increased as the speed increased (Table 1). Each time the stepper motor moved or stopped, it underwent stages of acceleration, constant motion, and deceleration. Measurement errors caused by the starting and stopping of the stepper motor can potentially be eliminated by removing the acceleration and deceleration stages. To this end, the signboards at the start and end points were moved by 0.6 m and 0.4 m toward the center of the sliding platform, respectively. As a result, the distance traveled by the LiDAR sensor was shortened to 4 m. Additionally, the measurement target was set close to the center of the sliding platform. The results showed that the errors presented in Table 2 were nearly eliminated. The remaining errors were equal to or smaller than the intervals between point-cloud datasets, and the systematic errors were caused by variations in the intervals of the LiDAR point cloud. With this alteration, the sliding platform was made suitable for canopy characterization.

### 3.2. Analysis of Results from the Simulated Canopy

The profiles, areas, and 3D canopy volumes obtained from the longitudinal and horizontal cross-sections measured at different positions using LiDAR and manual measurement were compared. The comparison results were used to determine whether the proposed CMPC method was viable.

The 3D canopy volumes obtained from the LiDAR and manual measurements are shown in Figure 8. It shows that these 3D canopy volumes were similar and only differ slightly in certain areas. This indicates that the proposed method is viable.

By analyzing the profiles obtained through LiDAR and manual measurements (Figure 9), it was found that the longitudinal and horizontal profiles captured by these methods for the bottom, middle, and top of the canopy were similar. The LiDAR-measured canopy profiles were generally larger than the manually measured ones. This is because the point-cloud-based canopy profile algorithm always selected the maximum canopy thickness in the vicinity of each grid space, which ensure that all the characteristics of the canopy can be obtained with no omission.

Figure 10 compares the areas of the longitudinal and horizontal cross-sections, which were measured at different heights of the canopy. The LiDAR and manually measured longitudinal and horizontal cross-sections correspond well at all positions. The differences between the plots were small. Consequently, the relative errors between the LiDAR and manual measurements are small. The LiDAR-based measurements of the simulated canopy volume achieved a maximum accuracy of 93.3% under a moving speed of 0.05 m/s, which is superior to the maximum accuracy achieved by ultrasonic sensors for regular orchard tree canopy profiles (92.8%) [18]. Therefore, it can be concluded that the present LiDAR-based canopy profile characterization method was suitable for measuring tree canopy volumes.

### 3.3. Results of CMPC Method

#### 3.3.1. 3D Point Cloud of Orchard Tree Canopy

The raw point-cloud data of the orchard tree was processed in MATLAB, as shown in Figure 11a. The left-hand side of the orchard tree was chosen as the primary subject of the study. To this end, a suitable range of x values was selected, and all point data except those representing the left-hand side of the target tree were excluded (Figure 11b).

#### 3.3.2. Extraction and Filtering of Lidar Point-Cloud Data

The point data corresponding to the side of the orchard tree facing the LiDAR platform were obtained by setting the appropriate canopy thickness (x) and height (z) values. The accepted x values were x > 0.64 m and x < 2 m. The distance between the outermost point of the orchard tree canopy and the center of the LiDAR sensor was 0.64 m, and the distance between the LiDAR sensor and the center of the orchard tree canopy cross-section was 2 m. Thus, all point-cloud data beyond the canopy center were excluded. The point cloud for half the canopy facing the sliding LiDAR platform was extracted (Figure 12). Regarding the z value range, all points above the bottom-most point of the canopy (z = 0.83 m) were extracted, and the *z*-axis coordinate was redefined (Figure 12). A coordinate transformation was performed to define the bottom-most point as the origin. Thus, all point data corresponding to the relevant portion of the target tree canopy were extracted, and the point cloud was stored as a .mat file. The canopy profile algorithm subsequent computations were performed by extracting data from this file.

#### 3.3.3. 2D and 3D Maps of Canopy

The CMPC method was used to process the point cloud of the orchard tree canopy and obtain the parameters. The “surf” and “images” functions were used to create 3D and 2D color maps of the canopy profile (Figure 13). The color bar represented the distribution of canopy volumes and thicknesses according to different numerical values in these maps, and the darker the color, the thinner was the canopy. Figure 13a,b showed that the LiDAR and manual measurements produced almost similar 3D shapes. Thus, the canopy shape of the orchard tree could be accurately obtained from LiDAR measurements. Comparing Figure 13c,d reveal that, the relative error between the LiDAR and manual measurements is seen to be small in the thicker (exceeding 0.8 m) parts of the canopy. These figures also indicate that the canopy is thinner at the edges and thicker at the center as expected.

#### 3.3.4. Comparison of CMPC Method and Manual Measurement Results 

By calculating the relative error between LiDAR detected and manually measured of canopy cross-section areas, the performance and effectiveness of the CMPC method for orchard tree detection were analyzed and verified. Comparison between LiDAR and manually measured longitudinal and horizontal (cross-section) canopy profiles.

Comparison between LiDAR and manually measured longitudinal and horizontal (cross-section) canopy profiles;

The horizontal cross-sections were extracted every 0.1 m along the longitudinal direction (*z*-axis) from the origin to the top of the canopy. Similarly, longitudinal cross-sections were extracted every 0.1 m along the *y*-axis. Figure 14 compared the profiles and cross-sectional areas of the LiDAR and manually measured longitudinal and horizontal cross-sections. The relative errors between these cross-sectional areas are shown in Table 3.

The results show that the LiDAR and manually measured longitudinal and horizontal cross-section profiles correspond well in the central positions (Figure 14 upper(b,c), lower(f,g)), whereas they corresponded poorly at the outer fringes (start and end points) of the canopy (Figure 14 upper(a,d), lower((e,h)). The poor correspondence between the LiDAR and manually measured canopy profiles at the fringe positions of the canopy may be caused by two factors: (1) The branches and leaves were sparse at the edges of the canopy resulting in the laser beam penetrating the canopy. The laser beam emitted by LiDAR was less intercepted. Therefore, the density of the point cloud was sparse. (2) The area of the receiving plate used in manual measurements (π × 0.052) was smaller than that of the grid (0.1 m × 0.1 m) in the CMPC method. The analysis of the cross-sectional areas in Figure 14 is presented in Table 3, and the results were consistent with the profile measurements. In positions where profile correspondence is poor, the relative errors in cross sectional area are large. Furthermore, the average relative error for the longitudinal cross-sectional area was 20%, smaller than the average relative error for horizontal areas 29% (note: cross-sections with anomalously large relative errors were excluded). Therefore, the measurement accuracy was higher in longitudinal cross-sections than in horizontal ones. The smallest relative error among the longitudinal cross-sections was 0.12%, which suggested a high level of accuracy in these measurements.

2.Comparison between LiDAR and manually measured areas and volumes for different cross-sections of the canopy.

Figure 15 compares the LiDAR and manually measured cross-sectional areas for longitudinal and horizontal cross-sections measured at different sections of the orchard tree′s canopy.

The trends depicted in Figure 15 indicate that the cross-sectional areas obtained by the CMPC method agree well with those obtained manually. Furthermore, it could be observed that the LiDAR-measured cross-sectional areas were larger than their manually measured counterparts, because LiDAR measurements of the canopy obtained points outlying the canopy. The differences between the LiDAR and manual measurements were smaller in the longitudinal cross-sections than in the horizontal ones, which was consistent with the analysis of their cross-sectional areas. This was because the LiDAR point cloud was obtained by scanning in the longitudinal direction and stitching in the horizontal direction, therefore the point-cloud series obtained from each scan corresponds to the longitudinal features of the scanned target. Thus, LiDAR-measured horizontal cross-sections were less accurate than LiDAR-measured longitudinal ones.

The relative errors between the CMPC and manually measured canopy volumes were analyzed. The manually measured canopy volume was found to be 1.61 m^3^, whereas the LiDAR-measured canopy volume (under a measuring speed of 0.05 m/s) was larger at 1.84 m^3^. The relative error was 14%. This is consistent with the LiDAR-measured canopy profile being larger than the manually measured canopy profile.

#### 3.3.5. Analyzing Effects of Moving Speed on Canopy Profile Characterization

Figure 16 shows the obtained profiles at different moving speeds. By analyzing the starting positions of the canopy profiles, positions, and directions (longitudinal/horizontal), no measurement lag in the LiDAR measurements were observed. The speed slightly influenced the measurement of the canopy profile, which was also observed by Liu et al. [41] and Shen et al. [42]. Since ultrasonic measurements always exhibit some degree of measurement lag, which increases with an increase in the moving speed [25], LiDAR measurements are superior in this respect.

At the central positions, the points position on the curves taken at different speeds were mostly the same, this made the curves superimposed. The correspondence between the LiDAR and manual measurement results was strong, and the relative errors in cross-sectional areas were comparatively small. At the beginning and end of the canopy, the relative errors of the cross-section area were large. The central position of the canopy was thicker than the outer fringes with a high point density. With the increase in detection speed, the cross-section area decreases gradually at the same position. The detection stability of the longitudinal profile was greater than that of transverse profile, because the slower the LiDAR detection speed, the higher the point cloud density. Moreover, the number of point cloud is in the unit grid. More detail of the canopy profile information was descripted. The canopy profile can be larger. On the contrary, the rougher of the canopy described, the canopy profile can be smaller.

The relative errors of the LiDAR-based canopy volume measurements obtained at different moving speeds are shown in Table 4.

In Table 4, the relative error (with respect to the measurement at 0.05 m/s) could be seen to increase with increasing moving speed, while the relative error of measured canopy volume with manual measurement gradually decreases. The measured canopy volume relative error to the manual measurement decreased with increasing moving speeds (2%, 2.7%, 2.5%, 1.2%, 1.3%, and 0.6%). The relative error implied that the LiDAR and manual measurements became more similar as the LiDAR point cloud became sparser. Compared to the canopy volume measurement taken at 0.05 m/s, the measurement error can be seen to increase with increasing moving speed. The relative errors with respect to LiDAR measurement taken at 0.05 m/s of these measurements increased by 2.2%, 2.1%, 2.2%, 1.1%, 1.1%, and 0.5%. The rate of increase in measurement error decreased as the moving speed increased.

A decrease in LiDAR moving speed increases the point-cloud density. As the representation of the canopy profile area became more detailed, the calculated canopy profile increased. Conversely, if it was made coarser, the canopy profile area would have decreased. The canopy volume exhibited the same trend as the profiles change. This showed that larger moving speeds produced smaller point-cloud densities and thereby reduced the quantity of information obtained by the LiDAR measurement. Thus, with increasing moving speed, the gap between the LiDAR-measured points was widened and the quantity of obtainable canopy information decreased. LiDAR measurement results approach those of the manual measurements. This means it is not necessary to process all the LiDAR point cloud data in the process of obtaining canopy volume by LiDAR information processing.

#### 3.3.6. Effects of Grid Size on Canopy Profile Characterization

Figure 17 compares the cross-section profiles obtained with different grid sizes for the same position. The 3D maps showed that the profile produced by the CMPC method became increasingly detailed as the grid size became smaller. However, the resulting cross-sectional profile also became increasingly discrete and uneven, making this method unsuitable for describing the canopy profile. Conversely, the CMPC method′s ability to describe the orchard tree profiles improves when the grid size was increased. Because the canopy profiles produced by this method were more realistic at large grid sizes. However, these profiles were less able to represent the details of the tree canopy. The CMPC method was suitable for describing the canopy structure with smaller gride sizes. Therefore, the CMPC method is simpler and more computationally efficient than previous LiDAR-based voxel methods for capturing orchard tree canopy structures [43,44,45].

Table 5 lists the canopy volumes, widths, and heights obtained using different grid sizes, which negligibly affected the 2D parameters (height and width) of the canopy. The effects of grid size on canopy volume were significant. The canopy volume obtained using the grid size of 0.01 m × 0.01 m was only 0.32 m^3^, which was approximately one-fifth of that calculated with a grid size of 0.1 m × 0.1 m. Therefore, the smallest grid size is unsuitable for variable pesticide application in orchards as it produces unrealistic canopy volumes. The ability of the algorithm to characterize the canopy volume improved with increasing grid sizes. At larger grid sizes, the calculated canopy volume approached the manually measured one. By comparing the results in Table 5, the grid size of the CMPC method should be set according to the measurement purpose. Small grid sizes were suitable for describing the finer structures of orchard tree canopies whereas large grid sizes are suitable for acquiring tree canopy profiles. The 2D parameters of the canopy can be accurately measured at all grid sizes.

## 4. Discussion

A high-precision detection method for canopy volume was designed for target-oriented spraying. The applicability of the CMPC method was evaluated by comparing the longitudinal and cross-section canopy profiles measured manually and by LiDAR. It was observed that the areas of the LiDAR-measured canopy profiles tend to be larger than manually measured ones. This is because the CMPC method always selects the largest values in the meshed area′s point cloud as the outermost points of the canopy. Consequently, the area of the manually measured canopy was slightly smaller than the point cloud obtained through contactless LiDAR measurements. This phenomenon also occurs in ultrasonic measurements of orchard-tree canopies [42]. The sparsity of the canopy and the limitations of ultrasonic signals (which suffer from increased measurement errors) mean that ultrasonic measurements also tend to overestimate canopy profile areas [21,23]. Overestimation of profile areas leads to inaccurate calculations of the canopy volume. However, the advantage was that appropriately increasing the dosage can ensure the control effect, and this is acceptable. Considering the overestimation of canopy profile, the advantages outweigh the disadvantages. The volumes calculated by CMPC were compared with ultrasonic measurements and the relative error of the LiDAR measured canopy volume was 14% in the orchard, while the ultrasonically measured relative error was 13% [21]. The measurement accuracies are similar, and CMPC is acceptable for variable spray. The sampling mode was the main contributor to the measurement error. Ultrasonically measured canopy profiles were similar to those measured manuals. The LiDAR sampling density in the horizontal direction was high, and the beams were longitudinally spaced to obtain longitudinal points. The point cloud was obtained from longitudinal scans that were distributed in the horizontal direction, differentiating CMPC from the manual method and give it application flexibility. Other sensors used for canopy detection such as the sensors of Kinect and Real Sense are not suitable for accurate canopy volume detection, because they have low sensing accuracy and are easily disturbed by the surrounding environment [11]. LiDAR is stable and has high accuracy. 

The speed of the moving LiDAR during measurement is one of the major factors affecting the performance of the CMPC method. The relative error between LiDAR and manual measurement results gradually decreased with increasing LiDAR moving speeds, and the error was within 3.7–14%. In low-speed detection, the LiDAR obtained point cloud was intensive. As the moving speeds increase the measured error becomes lower and the result gradually matches with manual measurement. This means that processing the entire point data is not necessary. A smaller point-cloud dataset can reduce the data processing requirements, save time, and improve measurement accuracy. Smaller LIDAR point cloud data have low computation and performance requirements for the controller, which are conducive to the popularization of this technology. Multiline LiDAR such as Velodyne, which can generate large point cloud data, is not necessary, and the 2D LiDAR is sufficient for determining the canopy characteristics using the CMPC method. The relative error between the 1 m/s and 1.2 m/s LiDAR measurements was 0.87%, acceptable for practical applications in agriculture. The relative errors at 1 m/s and 1.2 m/s speed measurements compared with manual measurement were 4.3% and 3.7%, respectively. These results confirm that LiDAR measurements are very close to manual measurements with relatively high accuracy. There was no detection lag for canopy profile detection with LiDAR at different moving speeds. In contrast, a small degree of measurement lag is present with ultrasonic measurements [25]. Laser propagates at the speed of light, so the LiDAR sensor can emit and receive laser signals quickly with no detection lag. Ultrasonic waves propagate at the speed of sound, so the detection lag exists. An orchard pesticide sprayer typically operates at speeds below 1.5 m/s [37]. The CMPC method can achieve an accuracy of 96.3% for orchard tree canopy volume measurements at a moving speed of 1.2 m/s. The calculation method in this study avoids the shortcomings of the previous studies, has high accuracy, has low computational requirements, and can applied to the variable spray process. On the other hand, LiDAR systems are expensive and their use in agriculture is a big investment for the farmer. Fortunately, the fast-paced development of LiDAR technology in agriculture may soon be economically feasible. Since the maximum speed of the sliding platform in this study was 1.2 m/s, the effects of moving speeds greater than 1.2 m/s would require further study.

## 5. Conclusions

In this work, an innovative CMPC method is proposed for calculating the volume of orchard tree canopies. The viability and accuracy of this method were verified by laboratory and field tests. The CMPC method achieved an accuracy of 93.3% when measuring a simulated canopy volume under a detection speed of 0.05 m/s. In the orchard tree canopy experiment, the accuracy of the CMPC method for canopy volume is 96.3% at a moving speed of 1.2 m/s. Therefore, the CMPC method is suitable for measuring orchard tree canopy volumes. The effects of LiDAR detection speed on orchard tree canopy characterization were investigated, and no measurement lag was found. The canopy volume measured by the CMPC method was largest when the LiDAR moving speed was 0.05 m/s. As the moving speed increased, the gap between the point-cloud datasets became larger and the CMPC method calculated volume approached to manually measured results. The relative errors of the canopy volume were less than 1% at moving speeds of 1.0–1.2 m/s, and measurement accuracy was essentially invariant within this range.

The effects of grid size on orchard tree canopy measurements in the CMPC method were investigated. Grid sizes between 0.01 m × 0.01 m to 0.05 m × 0.05 m were found to be suitable for describing the details of orchard tree canopies, and these details were easily extracted owing to the simplicity of the algorithm. Grid sizes between 0.075 m × 0.075 m and 0.1 m × 0.1 m were suitable for characterizing the profiles and volumes of orchard tree canopies and resulted in high measurement accuracies. Grid sizes of 0.1 m × 0.1 m or larger can be used for CMPC-method-based target-oriented spray application, and the grid size should be fine-tuned according to the number and height of sprayers on the spraying machine.

It is anticipated that the CMPC method can serve as the main technical reference for the development of target-oriented spray processes in the emerging field of intelligent agriculture. The proposed method can enable the effective and safe use of pesticides in orchard trees, increasing fruit yields while minimizing the contamination risks associated with agricultural activities.

## Figures and Tables

**Figure 1 sensors-21-04252-f001:**
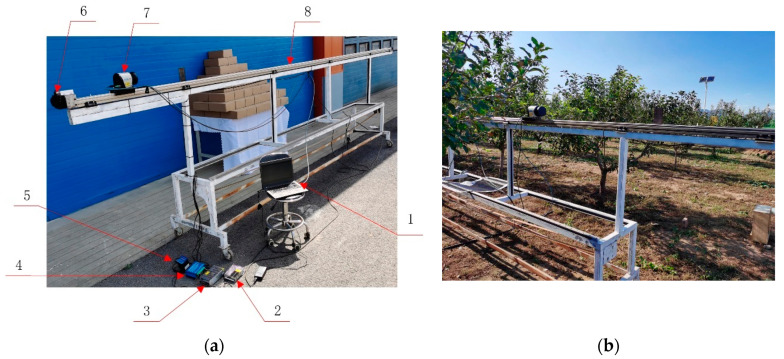
LiDAR system structure: (**a**) Sliding LiDAR system, including notebook computer (1), stepper motor driver power source (2), LiDAR sensor power source (3), stepper motor driver (4), stepper motor controller (5), stepper motor (6), LiDAR sensor (7), and linear rail guide (8); (**b**) LiDAR system canopy measurement setup.

**Figure 2 sensors-21-04252-f002:**
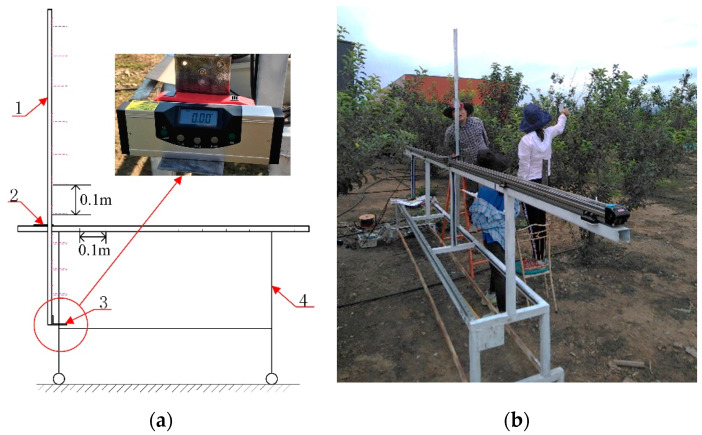
Manual measurement system: (**a**) Manual measurement system structure, including measuring instrument fixed bracket (1), slider (2), mounting plate for measuring instruments (3), sliding platform (4); (**b**) manual measurement process.

**Figure 3 sensors-21-04252-f003:**
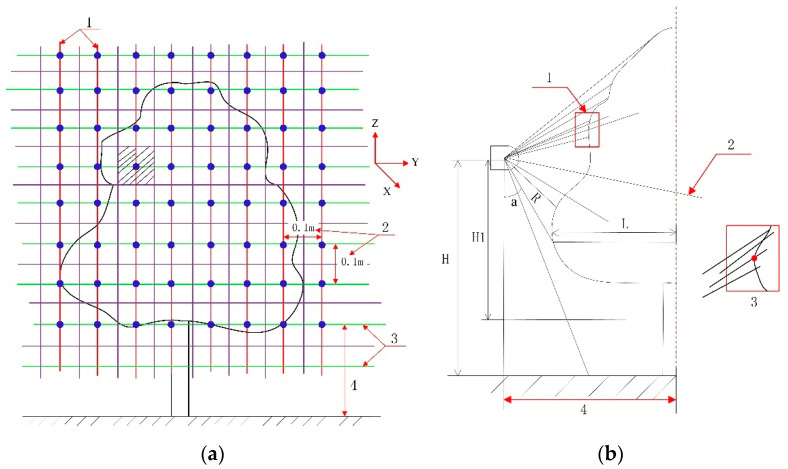
CMPC method: (**a**) Canopy profile mesh, including longitudinal profile extraction line (1), extraction spacing (2), horizontal profile extraction line (3), height from canopy bottom to the ground (4); (**b**) extraction of x_max_ from a meshed area, including fixed meshed area (1), a point beyond the tree trunk (2), extract of the outermost point of the longitudinal section of the canopy profile (3), horizontal distance between the center of the LiDAR system and canopy (4).

**Figure 4 sensors-21-04252-f004:**
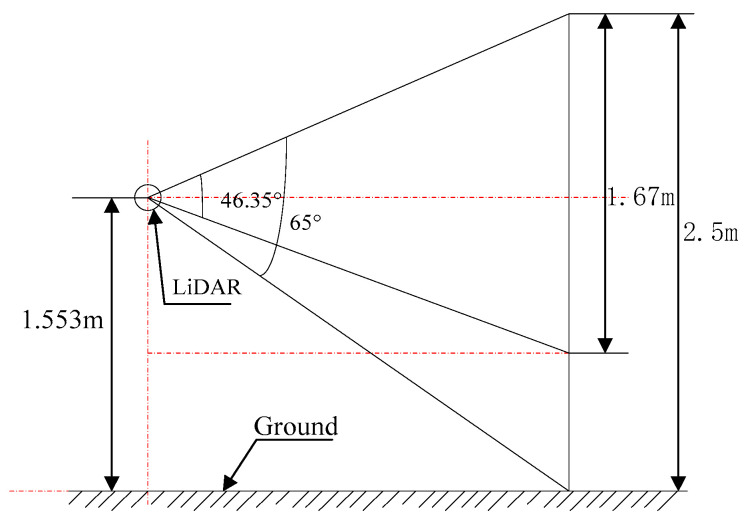
Illustration of canopy point-cloud calculations.

**Figure 5 sensors-21-04252-f005:**
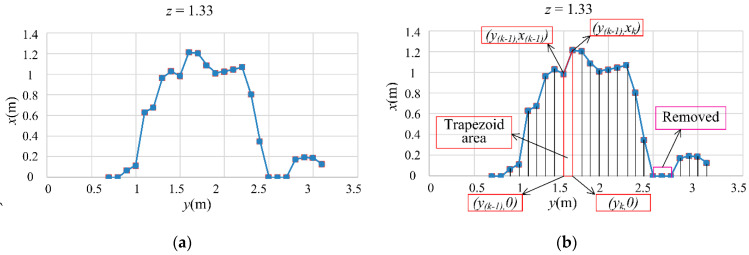
Calculation of the canopy′s cross-sectional areas; (**a**) Canopy′s cross-section; (**b**) Canopy′s cross-sectional areas calculation.

**Figure 6 sensors-21-04252-f006:**
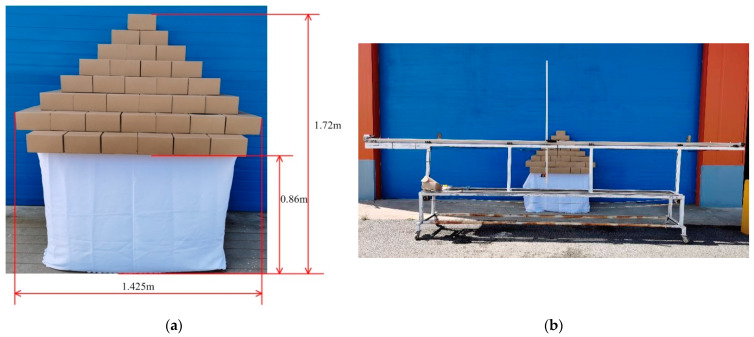
Simulated canopy and manual measurement system; (**a**) Simulated canopy dimensions; (**b**) Experimental study on simulated canopy detection.

**Figure 7 sensors-21-04252-f007:**
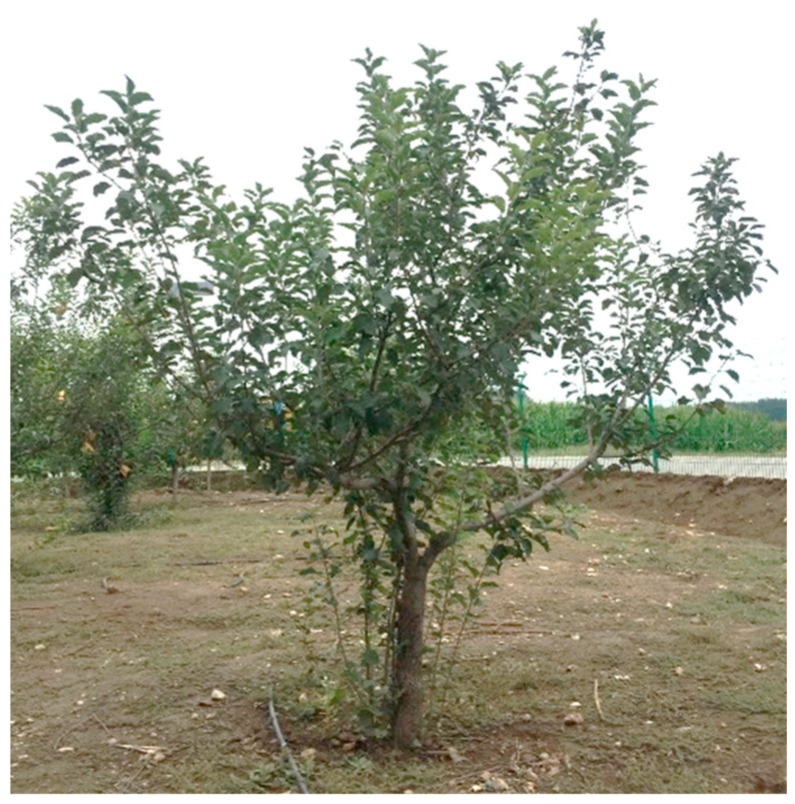
Target orchard tree.

**Figure 8 sensors-21-04252-f008:**
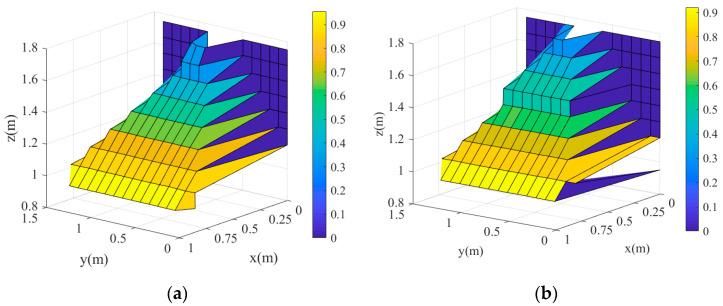
3D volumes of the simulated canopy: (**a**) LiDAR-measured 3D volume; (**b**) Manually measured 3D volume.

**Figure 9 sensors-21-04252-f009:**
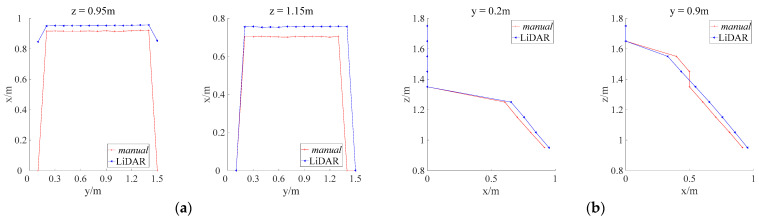
Comparison between LiDAR and manually measured longitudinal and horizontal cross-sections of the simulated canopy: (**a**) Horizontal sections of the canopy; (**b**) Longitudinal sections of the canopy.

**Figure 10 sensors-21-04252-f010:**
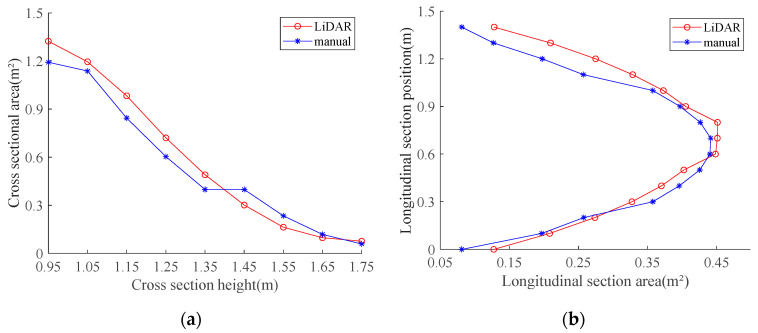
Comparison between LiDAR and manually measured horizontal and longitudinal cross-sectional areas at different positions for an orchard tree canopy: (**a**) Comparison between horizontal cross-section areas; (**b**) Comparison between longitudinal cross-section areas.

**Figure 11 sensors-21-04252-f011:**
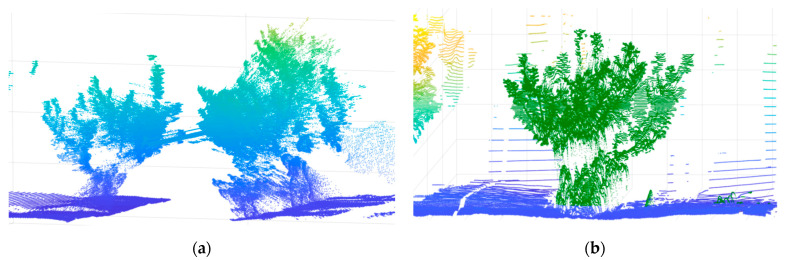
3D point cloud of orchard tree canopy: (**a**) LiDAR-measured 3D point cloud of orchard tree (both sides); (**b**) Point cloud of orchard tree (left-hand side).

**Figure 12 sensors-21-04252-f012:**
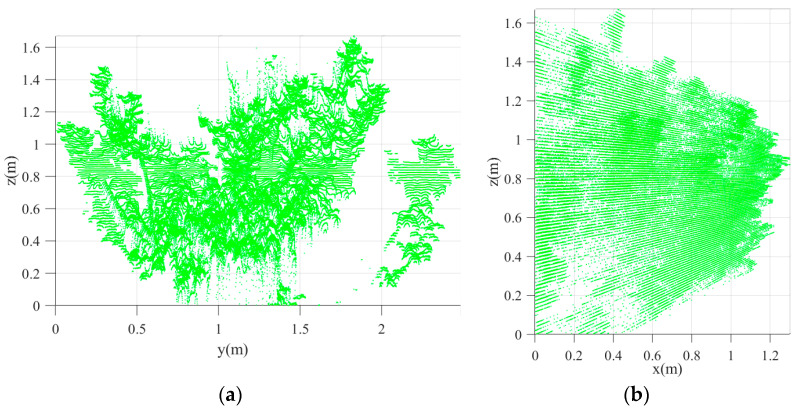
Point clouds of orchard tree canopy: (**a**) Front view; (**b**) Left view.

**Figure 13 sensors-21-04252-f013:**
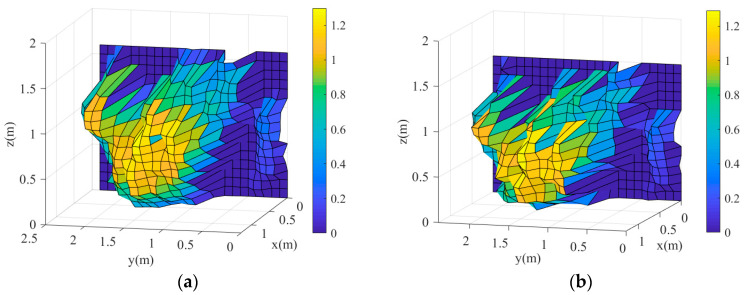
3D and 2D color maps of the canopy: (**a**) LiDAR-measured 3D map; (**b**) Manually measured 3D map; (**c**) LiDAR-measured 2D color map; (**d**) Manually measured 2D color map.

**Figure 14 sensors-21-04252-f014:**
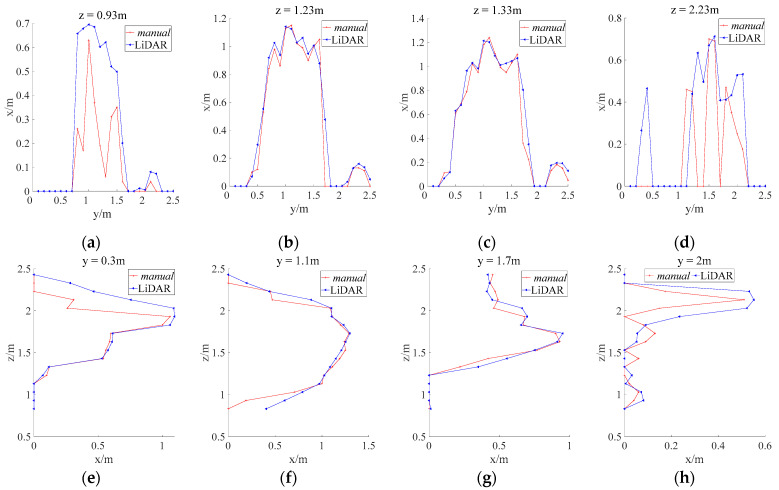
Horizontal (upper) and longitudinal (lower) cross-sections of the canopy: upper (**a**) Cross-section at the start of the canopy; upper (**b**) Cross-section in the middle of the canopy; upper (**c**) Cross-section in the middle of the canopy; upper (**d**) Cross-section at the end of the canopy; lower (**e**) Cross-section at the start of the canopy; lower (**f**) Cross-section in the middle of the canopy; lower (**g**) Cross-section in the middle of the canopy; lower (**h**) Cross-section at the end of the canopy.

**Figure 15 sensors-21-04252-f015:**
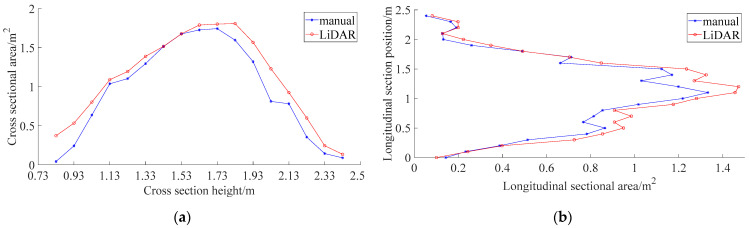
Comparison of LiDAR and manually measured cross-sectional areas in longitudinal and horizontal cross-sections measured at different positions in the orchard tree canopy: (**a**) Horizontal cross-sectional areas; (**b**) Longitudinal cross-sectional areas.

**Figure 16 sensors-21-04252-f016:**
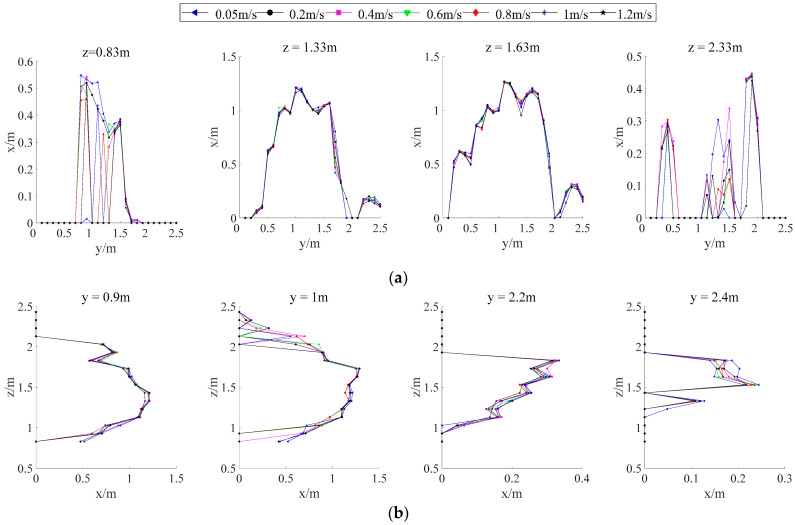
Comparison of longitudinal and horizontal canopy profile measurements obtained under different moving speeds: (**a**) Horizontal canopy profile measurements; (**b**) Longitudinal canopy profile measurements.

**Figure 17 sensors-21-04252-f017:**
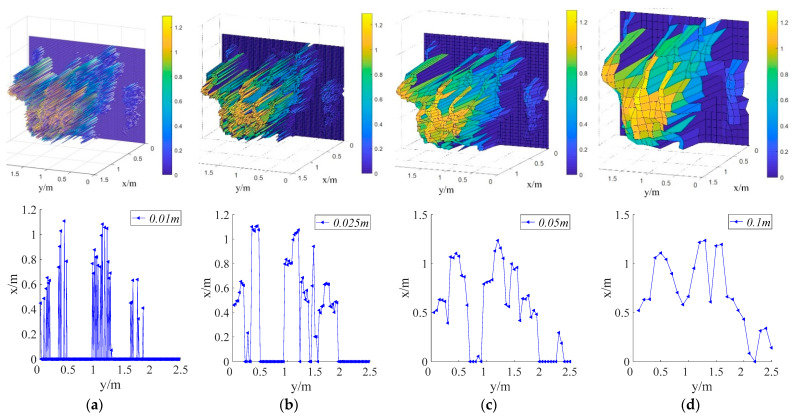
Comparison between 3D maps and cross-sectional orchard tree profiles obtained at different grid sizes: (**a**) 0.01 m × 0.01 m; (**b**) 0.025 m × 0.025 m; (**c**) 0.05 m × 0.05 m; (**d**) 0.1 m × 0.1 m.

**Table 1 sensors-21-04252-t001:** Errors of LiDAR sensor for the movement along the linear rail guide at each moving speed.

Moving Speed (M/S)	Distance Traveled (M)	Measurement Error (M)	Relative Error (%)
0.05	4.988	0.012	0.24
0.3	4.992	0.008	0.16
0.6	5.028	0.028	0.56
0.9	5.112	0.112	2.24
1.2	5.208	0.208	4.16

**Table 2 sensors-21-04252-t002:** Errors of modified LiDAR for the movement along the linear rail guide at each moving speed.

Moving Speed (M/S)	Distance Traveled (M)	Measurement Error (M)	Relative Error (%)
0.05	4.007	0.007	0.175
0.2	3.996	0.004	0.1
0.4	3.992	0.008	0.2
0.6	3.972	0.028	0.7
0.8	3.984	0.016	0.4
1.0	4	0	0
1.2	4.008	0.008	0.2

**Table 3 sensors-21-04252-t003:** Profile areas of horizontal and longitudinal cross-sections of the canopy.

Section	Position (M)	Manually Measured Area (M^2^)	Lidar Measured Area (M^2^)	Relative Error (%)
Horizontal cross-section	0.93	0.242	0.533	120
	1.23	1.104	1.194	8.2
	1.33	1.296	1.389	7.2
	2.23	0.355	0.599	69
Longitudinal cross-section	0.3	0.514	0.726	41
	1.1	1.332	1.455	9.2
	1.7	0.712	0.704	1.1
	2	0.132	0.223	69

**Table 4 sensors-21-04252-t004:** Relative errors of canopy volume measurements at different moving speeds.

Moving Speed (M/S)		Canopy Volume (M^3^)	Relative Error with Respect to Manual Measurement (%)	Relative Error with Respect to Lidar Measurement Taken at 0.05 M/S (%)
Manual measurement		1.61	0	12.5
LiDAR measurement	0.05	1.84	14	0
	0.2	1.80	12	2.2
	0.4	1.76	9.3	4.3
	0.6	1.72	6.8	6.5
	0.8	1.70	5.6	7.6
	1.0	1.68	4.3	8.7
	1.2	1.67	3.7	9.2

**Table 5 sensors-21-04252-t005:** Canopy volumes, widths, and heights measured using the manual method and CMPC method.

Measurement Method	Grid Size (M × M)	Volume (M^3^)	Canopy Width (M)	Canopy Height (M)
Manual	0.1 × 0.1	1.61	2.46	1.7
LiDAR measured speed 1 m/s	0.01 × 0.01	0.32	2.46	1.65
	0.025 × 0.025	1.02	2.46	1.65
	0.05 × 0.05	1.37	2.46	1.65
	0.075 × 0.075	1.55	2.46	1.65
	0.1 × 0.1	1.68	2.46	1.65

## Data Availability

Not applicable.

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
