# Peer review of "CMPC: An Innovative Lidar-Based Method to Estimate Tree Canopy Meshing-Profile Volumes for Orchard Target-Oriented Spray"

_sensors, 2021, doi:10.3390/s21124252_

Round 1
Reviewer 1 Report
This manuscript focuses on a novel canopy meshing-profile characterization (CMPC) method based on light detection and ranging (LiDAR)point-cloud data was designed for high-precision canopy volume calculates is presented in this article. For the validation of this method, tests have been developed in the laboratory and in the field in order to validate the measured data.
It is relatively clear what new scientific methods insights are gained from this manuscript given works have been done in this area. The paper is well written, structured and cited. Abstract, Introduction and Methods are well written and clear. Some sentences and “tipos” must be attendant.
There are some comments about the article:
-A better detailed must be given, about how the speed ranges were established for the displacement of the System: 1 m / s and 1.2 27 m / s
-Figure 16 a and b should be improved, since the superimposed curves are not well visualized
-A table should be adjusted 5. Within these data, the exact manual measurement of the initial volume should be established in detail.
-Some references of relevant related works within the field of point cloud generation and processing should be included as: Robotic Fertilisation Using Localisation Systems Based on Point Clouds in Strip-Cropping Fields (https://www.mdpi.com/2073-4395/11/1/11)
-Interesting article. The benefits and relevant aspects of the proposed method should be included in the discussion compared to similar datasets or other sensors capable of generating point clouds such as: Velodyne (3D), Real Sense, Kinect.
-Why has Matlab been used instead of a real-time system like ROS? Which is more specialized in the processing of point clouds and this type of laser
Author Response
Dear Reviewer,
On behalf of all the authors, we sincerely appreciate your valuable comments on the manuscript. Your comments not only provide constructive suggestions on improving the quality of the manuscript but also lead us to consider our approaches and the design of the systems in detail. Our future research will benefit from these comments, as well.
Best regards,
Chenchen Gu, Changyuan Zhai, Xiu Wang and Songlin Wang
Point 1: A better detailed must be given, about how the speed ranges were established for the displacement of the System: 1 m / s and 1.2 27 m / s
Author response: Thank you for the suggestion. We updated the manuscript by adding explain of the speed ranges establishment. (Yellow highlight in L127-129)
Point 2: Figure 16 a and b should be improved, since the superimposed curves are not well visualized.
Author response: A sufficient explanation is added about the figure 16 for the superimposed curves. (Yellow highlight in L430-431)
Point 3: A table should be adjusted 5. Within these data, the exact manual measurement of the initial volume should be established in detail.
Author response: The number of Table 5 was wrong defined which should be table 4, and the number of Table 6 was modified as table5. The manual measurement of the initial volume was added in table 4. The relative errors were calculated and show. (Yellow highlight in L442-444 and 483)
Point 4: Some references of relevant related works within the field of point cloud generation and processing should be included as: Robotic Fertilisation Using Localisation Systems Based on Point Clouds in Strip-Cropping Fields (https://www.mdpi.com/2073-4395/11/1/11)
Author response: The suggestion is very good. We updated the references. (Yellow highlight in L675-676)
Point 5: Interesting article. The benefits and relevant aspects of the proposed method should be included in the discussion compared to similar datasets or other sensors capable of generating point clouds such as: Velodyne (3D), Real Sense, Kinect.
Author response: The discussion was added. (Yellow highlight in L521-524 and L534-536)
Point 6: Why has Matlab been used instead of a real-time system like ROS? Which is more specialized in the processing of point clouds and this type of laser
Author response: Thank you for the suggestion. MATLAB is a powerful tool for data processing and development, with the advantages of easy to learn, powerful processing function and modular tools. So, this study applied MATLAB to realize and verify the feasibility of CMPC algorithm. Our team is developing ROS system. We hope the ROS work well for orchard

Reviewer 2 Report
I found the study to be well designed and implemented, and the manuscript does an excellent job of presenting the study data and results. I suggest a minor English grammar and style edit to improve the quality of the manuscript from good to excellent. For example on line 16, "canopy volume calculates" should be revised to "canopy volume calculations." There are similar errors in English grammar throughout the manuscript.
Also, I suggest the authors revise sentences such as lines 24-25, ""The cross-section profiles can fit with the manually measured results very well." What does "very well" mean? This is an example of poor scientific writing. There are others throughout the manuscript.
An English speaking editor should be able to revise the errors quickly.
Author Response
Point 1: I found the study to be well designed and implemented, and the manuscript does an excellent job of presenting the study data and results. I suggest a minor English grammar and style edit to improve the quality of the manuscript from good to excellent. For example, on line 16, "canopy volume calculates" should be revised to "canopy volume calculations." There are similar errors in English grammar throughout the manuscript.
Author response: Thank you for the suggestion. The manuscript was revised as suggested. (Yellow highlight in L16 and L213)
Point 2: Also, I suggest the authors revise sentences such as lines 24-25, ""The cross-section profiles can fit with the manually measured results very well." What does "very well" mean? This is an example of poor scientific writing. There are others throughout the manuscript.
Author response: Thank you for the suggestion. The manuscript was revised. (Yellow highlight in L23-24, L297-298, L301-302, L352-354, L539-541)
Point 3: An English speaking editor should be able to revise the errors quickly.
Author response: The manuscript had been polished by an English-speaking editor.

Reviewer 3 Report
The paper aims to develop a new method for high-precision characterization of tree canopy volume using terrestrial lidar data for agricultural applications. The text is well written and easy to understand, although some sentences have problems with wrong present/past tense. I cannot confirm about paper novelty because I do not have knowledge about this specific application of lidar. However, it is scientifically sound. Since I do not have any major objections to the paper, I recommend minor revisions and offer the authors some general and specific comments to improve the paper:
1) Introduction: I am not familiar with pesticide application in orchards, thus I do not understand exactly how the target-oriented spray would work. Would a machine carrying the lidar/ultrasonic sensor measure the canopy volume and proceed to apply the correct dosage of the pesticide? This may be obvious for someone in the field but perhaps adding one sentence in the Introduction to clarify this would be important for general readership.
2) Methods/Results: there is some mixture of Methods in Results, so I suggest authors to revise Results thoroughly moving anything that is not reporting a result to Methods - one example is the Volume equation.
3) Some minor English revision should be done.
Specific comments:
L45, I would expect also to reduce costs of production since you use less pesticides, perhaps this could be important to mention
L48-50, including some examples of common methods/data besides the cited most popular ones could be interesting
L89, represented -> represented
L91, “were included” why use past here? I do not understand, revise these sentences. Also, should probably cite some studies to support the use of these metrics
L100, I would suggest placing the objective in a new paragraph for when the reader wants to go directly to objective it is easier to find
L104, wouldn’t be more suitable to say field experiments instead of ‘orchard experiments’? if you agree please take a look throughout the paper and correct because there are more instances of this
L107-108, for the “application of” ? the target-oriented spray repeats in this sentence, revise
L123, why 1.2 m/s? I am not familiar with pesticide application but is this the speed that a machine moves in the field?
L201, define the HSK term in the text
L392, this is Methods, move to the appropriate section
L455, this is also Methods, describe here only the results, e.g. which grid has better results?
L530, this information should be in Methods justifying the tested speed
L536-537, revise this sentence for English, needs to be better flowing
L543, remove ‘can’
Author Response
Dear Reviewer,
On behalf of all the authors, we sincerely appreciate your valuable comments on the manuscript. Your comments not only provide constructive suggestions on improving the quality of the manuscript but also lead us to consider our approaches and the design of the systems in detail. Our future research will benefit from these comments, as well.
Best regards,
Chenchen Gu, Changyuan Zhai, Xiu Wang and Songlin Wang
The paper aims to develop a new method for high-precision characterization of tree canopy volume using terrestrial lidar data for agricultural applications. The text is well written and easy to understand, although some sentences have problems with wrong present/past tense. I cannot confirm about paper novelty because I do not have knowledge about this specific application of lidar. However, it is scientifically sound. Since I do not have any major objections to the paper, I recommend minor revisions and offer the authors some general and specific comments to improve the paper:
Point 1: Introduction: I am not familiar with pesticide application in orchards; thus, I do not understand exactly how the target-oriented spray would work. Would a machine carrying the lidar/ultrasonic sensor measure the canopy volume and proceed to apply the correct dosage of the pesticide? This may be obvious for someone in the field but perhaps adding one sentence in the Introduction to clarify this would be important for general readership.
Author response: Thank you for the suggestion. The manuscript was revised as suggested. (Yellow highlight in L44-46)
Point 2: Methods/Results: there is some mixture of Methods in Results, so I suggest authors to revise Results thoroughly moving anything that is not reporting a result to Methods - one example is the Volume equation.
Author response: The suggestions made by the reviewer are very good. We updated the manuscript as suggested. (Yellow highlight in L213-215 and L261-266)
Point 3: Some minor English revision should be done.
Author response: The manuscript had been polished by an English-speaking editor.
Specific comments:
Point 4: L45, I would expect also to reduce costs of production since you use less pesticides, perhaps this could be important to mention
Author response: The manuscript was revised as suggested. (Yellow highlight in L46-47)
Point 5: L48-50, including some examples of common methods/data besides the cited most popular ones could be interesting
Author response: We revised the manuscript as suggested. (Yellow highlight in L50-56)
Point 6: L89, represented -> represented?
Author response: We revised this sentence. (Yellow highlight in L96)
Point 7: L91, “were included” why use past here? I do not understand, revise these sentences. Also, should probably cite some studies to support the use of these metrics
Author response: The part was modified. (Yellow highlight in L97-98)
Point 8: L100, I would suggest placing the objective in a new paragraph for when the reader wants to go directly to objective it is easier to find
Author response: The manuscript was revised as suggested. (Yellow highlight in L106)
Point 9: L104, wouldn’t be more suitable to say field experiments instead of ‘orchard experiments’ ? if you agree please take a look throughout the paper and correct because there are more instances of this
Author response: The manuscript was revised as suggested. (Yellow highlight inL20, L109, L117, L238)
Point 10: L107-108, for the “application of”? the target-oriented spray repeats in this sentence, revise
Author response: The part was modified. (Yellow highlight in L112-113)
Point 11: L123, why 1.2 m/s? I am not familiar with pesticide application but is this the speed that a machine moves in the field?
Author response: The selection criteria is added. (Yellow highlight in L127-129)
Point 12: L201, define the HSK term in the text
Author response: HSK means Horizontal section area, but I found this is a wrong definition of the profile area. We updated the manuscript by redefining Horizontal section area, and made a modify of the formula 1. (Yellow highlight in L208-210)
Point 12: L392, this is Methods, move to the appropriate section
Author response: The manuscript was revised as suggested. (Yellow highlight in L213-215)
Point 13: L455, this is also Methods, describe here only the results, e.g. which grid has better results?
Author response: The manuscript was revised as suggested. (Yellow highlight in L261-266)
Point 14: L530, this information should be in Methods justifying the tested speed
Author response: The part was modified. (Yellow highlight in L127-129)
Point 15: L536-537, revise this sentence for English, needs to be better flowing
Author response: The part was revised as suggested. (Yellow highlight in L551-552)
Point 16: L543, remove ‘can’
Author response: The part was revised as suggested. (Yellow highlight in L558-559)
Round 2
Reviewer 1 Report
The authors have made the suggested changes.